# The Emerging Role of Mitochondrial Dysfunction in Thyroid Cancer: Mediating Tumor Progression, Drug Resistance, and Reshaping of the Immune Microenvironment

**DOI:** 10.3390/biom15091292

**Published:** 2025-09-08

**Authors:** Yating Zhang, Hengtong Han, Tingting Zhang, Tianying Zhang, Libin Ma, Ze Yang, Yongxun Zhao

**Affiliations:** 1The First School of Clinical Medicine, Lanzhou University, Lanzhou 730030, China; zhangyt2024@lzu.edu.cn (Y.Z.); zhangtt2024@lzu.edu.cn (T.Z.); zhangty2024@lzu.edu.cn (T.Z.); 2The Seventh Department of General Surgery, Department of Thyroid Surgery, The First Hospital of Lanzhou University, Lanzhou 730030, China; hanht20@126.com (H.H.); malb2012@126.com (L.M.); doccrazy11@163.com (Z.Y.)

**Keywords:** thyroid cancer, mitochondria, mitochondrial dynamics, mitochondrial autophagy, mitochondrial metabolism

## Abstract

As the hub of energy metabolism and the cell’s fate arbiter, mitochondria are essential for preserving cellular homeostasis and converting it from pathological states. Therefore, through mechanisms that drive metabolic reprogramming, oxidative stress, and apoptosis resistance, mitochondrial dysfunction (including mitochondrial DNA mutations, mitochondrial dynamics imbalance, mitochondrial autophagy abnormalities, mitochondrial permeability abnormalities, and metabolic disorder) can promote the progression of thyroid cancer (TC), resistance to treatment, and reshaping of the immune microenvironment. This article reviews the molecular mechanisms and characteristic manifestations of mitochondrial dysfunction in TC. It focuses on providing a summary of the main strategies currently used to target the mitochondria, such as dietary intervention and targeted medications like curcumin, as well as the clinical translational value of these medications when used in conjunction with current targeted therapies for TC and radioactive iodine (RAI) therapy in patients with advanced or RAI-refractory TC who rely on targeted therapies. The application prospects and existing challenges of emerging therapeutic methods, such as mitochondrial transplantation, are also discussed in depth, aiming to provide new perspectives for revealing the molecular mechanisms by which mitochondrial dysfunction drives the progression of TC, drug resistance, and the reshaping of its immune microenvironment, as well as providing new diagnostic and therapeutic strategies for patients with advanced or RAI-refractory TC who are reliant on targeted therapies.

## 1. Introduction

Mitochondria, as the central hub of cellular metabolic regulation, orchestrate energy production, biosynthesis, and signal transduction through oxidative phosphorylation (OXPHOS), metabolic reprogramming, and interorganellar communication, playing a pivotal role in cell fate determination and immunometabolic regulation [1,2]. However, disturbances in the mitochondrial quality control (MQC) system may lead to mitochondrial dysfunction, which subsequently contributes to tumor progression, chemoresistance, and the reshaping of the tumor microenvironment in various malignant tumors, including lung cancer, hepatocellular cancer, pancreatic cancer, and breast cancer [3]. Specifically, as lung cancer remains the leading cause of cancer-related deaths in the United States, research by Subhadeep Das and colleagues demonstrated that the anticancer compound supinoxin suppresses small-cell lung cancer proliferation and tumor growth by targeting DDX5 to inhibit mitochondrial gene expression and disrupt mitochondrial respiration, thereby depriving tumor cells of energy and exerting an antitumor effect [4,5]. In hepatocellular cancer, the tumor microenvironment causes mitochondrial dysfunction, which encourages tumor growth and resistance to chemotherapy. The concrete manifestations include increased oxidative stress, altered mitochondrial dynamics, impaired mitochondrial autophagy, and the initiation of new cell death pathways, such as cupric-mediated apoptosis and ferroptosis [6]. Mitochondrial dysfunction through metabolic adaptation alterations and apoptosis inhibition, which promotes treatment resistance in pancreatic cancer [7], while in breast cancer, drives progression and treatment resistance by reshaping the microenvironment [8]. According to research, ellagic acid targets mitochondria and lowers the mitochondrial membrane potential, thereby inhibiting tumor cell proliferation and inducing apoptosis. This has anticancer effects on breast, pancreatic, and hepatocellular cancers [9]. Consequently, mitochondrial dysfunction has gradually emerged as both a novel therapeutic target for cancer treatment and a critical breakthrough point for reversing drug resistance.

TC ranks as the most prevalent malignancy within the endocrine system. Based on information from the GLOBOCAN 2022 database of the International Agency for Research on Cancer, there were 47,507 TC-related deaths and an anticipated 821,214 new cases of TC diagnosed worldwide in 2022. With 56.77% of all cases and 24.35% of all TC-related deaths worldwide, China leads in both TC incidence and TC-related mortality among 185 nations, exhibiting a steady upward trend [10,11]. TC is primarily divided into four subgroups based on histopathologic characteristics: papillary thyroid cancer (PTC), follicular thyroid cancer (FTC), medullary thyroid cancer (MTC), and anaplastic thyroid cancer (ATC). Among them, ATC is formed through de-differentiation and represents the most aggressive and de-differentiated subtype of TC [12,13,14]. TC treatment has entered the era of precision medicine, and while a comprehensive surgical-based treatment strategy for PTC has a favorable overall prognosis, with a five-year survival rate of up to 97%, treating locally advanced, RAI-refractory PTC, MTC, and ATC, with a five-year survival rate of only 39%, remains a clinical challenge [15]. Even though the aforementioned TC subtypes have shown progress with targeted therapeutic approaches, they still have trouble with low clinical response rates and short-term targeted therapy resistance [16]. As a result, an in-depth study of molecular mechanisms, including etiology and targeted therapeutic resistance, is required for various TC subtypes. Recent research has shown that mitochondrial dysfunction, including mitochondrial DNA (mtDNA) mutations and metabolic reprogramming, activates pro-inflammatory and oxidative stress pathways, which in turn promote treatment resistance, metastasis, and malignant proliferation of TC, particularly MTC and ATC [17,18,19,20,21].

The molecular mechanism of mtDNA mutations, mitochondrial dynamics abnormalities, mitochondrial autophagy abnormalities, mitochondrial permeability alterations, and metabolic reprogramming in the development of TC are systematically reviewed in this paper. We also have in-depth discussions about the potential clinical application value of novel therapeutic approaches based on metabolic interventions (such as fasting-mimicking diets), small-molecule-targeted medications (such as Drp-1 inhibitors), mitochondrial transplantation, and combination therapies. This article aims to elucidate the molecular mechanisms by which mitochondrial dysfunction contributes to the development of TC and to provide innovative intervention strategies for clinical treatment.

## 2. Key Concepts and Review Methodology

### 2.1. Key Concepts

OXPHOS. OXPHOS is the process by which cells use oxygen and carbon fuels to produce ATP inside their mitochondria. Conventional wisdom maintains that cancer cells undervalue the function of OXPHOS by favoring glycolysis (the Warburg effect) [22]. New research, however, reveals that the degree of glycolytic reliance varies across TC subtypes. As the primary bioenergetic pathway in PTC, OXPHOS, for example, not only stays uninhibited but also promotes PTC growth, survival, and treatment resistance by providing an effective energy source and biosynthetic precursors [19,23].

Reactive oxygen species (ROS). Superoxide anion, hydroxyl radical, hydrogen peroxide, and other highly reactive oxygen-derived molecules are together referred to as ROS. They can be produced by external stimuli or NADPH oxidase, but their primary source is electron leakage in complexes I and III of mitochondrial energy metabolism [24]. ROS have multiple biological roles in malignancies, acting as both pro-oncogenic and anti-oncogenic substances. Antioxidant systems can buffer moderate increases in ROS, preserving pro-oncogenic redox balance to promote tumor growth; on the other hand, excessive ROS accumulation causes cancer cells to undergo apoptosis [25].

Epithelial–mesenchymal transition (EMT). A vital biological process known as the EMT causes cancer cells to transition from an epithelial to a mesenchymal phenotype. EMT promotes tumor growth and treatment resistance by increasing cellular plasticity, anti-apoptotic ability, and metastatic potential [26]. According to research, the “ROS-glycobiology-EMT” axis posits that ROS influence the EMT process and accelerate the progression of cancer by modulating glycosyltransferases, thereby altering protein glycosylation (glycan structure) [27].

### 2.2. Review Methodology

This study carefully retrieved the scholarly literature on the relationship between mitochondrial dysfunction and TC using the PubMed database and the literature management program Zotero 7.0.9 (Corporation for Digital Scholarship, Fairfax, United States). The following keywords and their combinations were used extensively in the search strategy: “TC,” “mitochondrial dysfunction,” “mt DNA,” “mitochondrial metabolism,” “mitochondrial dynamics,” “mitochondrial permeability,” “mitochondrial autophagy,” and “mitochondria-targeted remedy.” The literature search work aims to discuss the pathogenesis of TC’s mitochondrial dysfunction and create targeted treatment plans. We prioritized peer-reviewed English-language articles that examined this mechanism, including review papers and in vitro/in vivo experimental studies, among the 498 documents initially retrieved. We included the most recent research findings up to 2026. We excluded papers that had little bearing on the topic of mitochondrial dysfunction in TC, as well as editorials, letters, conference abstracts, and articles written in languages other than English. In the end, approximately 200 studies were chosen for inclusion, with a particular emphasis on original research papers and systematic reviews published in the last ten years, to ensure that the integrated evidence is both up-to-date and comprehensive.

## 3. Processes and Mechanisms of Mitochondrial Homeostasis Regulation

As the core of cellular energy metabolism and homeostatic regulation, mitochondria participate in numerous biological processes, including the production of ATP, signaling, protein synthesis, DNA stability, and the MQC system, which maintains mitochondrial homeostasis [2]. To ensure that the energy supply meets the cell’s demands and to eliminate damaged components, the cell employs complex regulatory mechanisms to maintain the amount, structure, function, and quality of mitochondria in a state of dynamic equilibrium, known as mitochondrial homeostasis. Sustaining cellular energy metabolism and calcium balance, scavenging damage components, and modulating immunological and death signals all depend on mitochondrial homeostasis [28]. Specifically, by coordinating the three processes of mitochondrial biosynthesis, mitochondrial dynamics (mitochondrial fusion and fission), and mitochondrial autophagy, MQC can create a sophisticated and dynamic mitochondrial homeostatic regulation network [29,30,31,32]. For instance, mitochondria may clear out damaged, senescent mitochondria through mitochondrial autophagy and mitochondrial fission, and they can produce new mitochondria through biogenesis [33,34]. Additionally, mitochondrial fusion can facilitate the exchange of mitochondrial contents, such as proteins and mtDNA, and “compensate” for damage to the mitochondria [35].

Nevertheless, mitochondrial homeostasis will be disrupted and mitochondrial dysfunction will be triggered when intracellular or extracellular stress-induced mitochondrial damage surpasses the MQC’s ability to repair it or when vital MQC components (such as the PINK1/Parkin-mediated mitochondrial autophagy pathway) themselves experience functional failure as a result of genetic defects or prolonged stress [33]. There are two types of mitochondrial dysfunction: acquired (caused by mutations in mtDNA) and secondary (caused by abnormalities in mitochondrial dynamics, dysregulated mitochondrial autophagy, altered permeability, and metabolic disturbances). Notably, these mitochondrial dysfunctions can interact with and influence one another [36]. For instance, excessive creation of ROS, calcium overload, and mitochondrial metabolic reprogramming may all exacerbate mitochondrial dysfunction, creating a vicious cycle [23]. Not only do the aforementioned mitochondrial pathological processes cause apoptosis and significantly disrupt cellular energy metabolism and signaling, but they can also create a mitochondrial oncogenic network that promotes TC and other cancers to progress and become resistant to targeted treatments. Specifically, by lowering OXPHOS, raising ROS, and encouraging EMT, mitochondrial dysfunction encourages tumor proliferation and invasion. Tumor recurrence and medication resistance are tightly linked to abnormal mitochondrial dynamics (such as upregulated OPA-1 expression) and metabolic reprogramming. Furthermore, inflammatory responses triggered by mitochondrial dysfunction may impair the immune system, allowing cancer cells to evade immune clearance and ultimately promoting metastasis [37]. The function of immunological cells to perform immunological surveillance is impaired by mitochondrial dysfunction, which further contributes to immune escape from tumors [38]. According to the aforementioned findings, mitochondrial dysfunction plays a crucial part in the development of several cancers, including TC. Additional research into the molecular mechanisms and complex regulatory networks underlying mitochondrial dysfunction in cancer will provide a theoretical foundation for developing innovative mitochondrial-targeted therapies to treat malignancies and overcome treatment resistance.

## 4. Mitochondrial Dysfunction in TC

The connection between the onset of TC and mitochondrial dysfunction has drawn a lot of attention in recent years [21]. Genetic factors (such as the *BRAF V600E/RAS* mutation), environmental exposures (such as radiation), metabolic disorders, and aging can all cause TC mitochondrial dysfunction (Figure 1) [39]. Through a vicious loop of “metabolic reprogramming—ROS burst—genomic damage,” mitochondrial dysfunction accelerates the course of TC [40,41,42]. The pathogenic pathways of TC mitochondrial dysfunction will be deeply investigated in this section, along with potential targeted therapeutics based on mitochondrial dysfunction targets. This will offer novel opportunities on the etiology, diagnosis, and management of TC (Figure 2; Table 1).

### 4.1. MtDNA Mutations: The Saboteur of TC

MtDNA, serving as the core genetic carrier for cellular energetics, exhibits a substantially higher mutation rate compared to nuclear DNA. According to conventional views, ROS is the primary cause of mtDNA mutations. However, new research has revealed that mtDNA mutations in tumor cells display a strand asymmetric distribution feature, indicating that replication errors might be a more important mutagenesis mechanism [55,56]. Anna L. M. Smith et al. found that mtDNA mutations keep accumulating as people age [57], causing a vicious cycle of “mtDNA mutation-ROS burst-mtDNA oxidative damage” by compromising the OXPHOS system [58,59], which fosters the growth and development of tumors. Notably, in PTC tall cell variant, one study found a strong association between mtDNA mutations and *BRAF V600E* driver mutations [44]. However, it remains to be confirmed whether this link holds for all *BRAF*-mutant PTC, and it is debatable whether mtDNA mutations accumulate passively or work in concert with *BRAF V600E*.

Recent studies generally agree that selective pressures maintaining mitochondrial function exist in cancer. Some researchers propose that mtDNA mutations are predominantly “passenger alterations,” while others suggest they are primarily “driver mutations.” However, there is currently nothing in the literature supporting any of the aforementioned patterns which predominantly characterize mtDNA mutations in TC [55,60,61]. But in PTC, studies have discovered that mtDNA mutations can contribute to invasion, metastasis, and cellular proliferation in several ways: (1) impairing OXPHOS function [44]; (2) promoting metabolic reprogramming [45]; (3) raising ROS generation [44]; and (4) facilitating genomic instability [43]. According to data analyzed by Khaled K. Abu-Amero et al., mtDNA mutations in PTC cells reached 36.8%, whereas mtDNA mutations in cells with benign thyroid hyperplasia might reach 25%. They found that, by altering energy metabolism and raising oxidative stress, mtDNA mutations encourage the development of PTC and FTC. They also found mtDNA mutations in benign hyperplasia, indicating that mtDNA mutations may be involved in the early stage of TC development [62]. Subsequently, it was discovered that the *m.3842G>A* mutation in the mtDNA *MT-ND1* gene, which was discovered to be present in TC, activates the ERK1/2 signaling pathway, significantly reducing OXPHOS function, and raises ROS levels which, taken together, promotes TC emergence, proliferation, invasion, and metastasis [63]. Notably, Xingyun Su et al. discovered that patients with PTC had aberrant alterations in their mtDNA copy number (mtDNA-CN) [64]. Subsequently, research by Materah Salem Alwehaidah et al. further confirmed that PTC patients had a significantly higher relative mtDNA-CN (2.1 ± 0.8) than the healthy control group (0.9 ± 0.4). This indicates a significant positive correlation between elevated mtDNA-CN and the risk of developing PTC [65], suggesting that elevated mtDNA-CN could be a biomarker for detecting early PTC and predicting the risk of developing PTC. In summary, mtDNA abnormalities are strongly linked to the occurrence, proliferation, invasion, and metastasis of TC. An in-depth analysis of mtDNA variation characteristics in TC cells will not only provide new theoretical foundations for elucidating TC etiology but also contribute to TC evaluation and the development of precision treatment strategies.

### 4.2. Mitochondrial Dynamics: A Lever for TC Cells’ Fate

Via the processes of mitochondrial fission, mitochondrial fusion, mitochondrial autophagy, and transport, mitochondrial dynamics accurately regulate and control the morphology and function of the mitochondria and are essential for energy metabolism, signaling, and MQC [66,67,68,69]. Under pathological conditions, oxidative stress and calcium dyshomeostasis disrupt the balance of mitochondrial dynamics-related proteins—characterized by aberrant activation of fission proteins (such as Drp1) and suppressed expression of fusion proteins (such as Mfn2)—leading to excessive mitochondrial fragmentation and cristae structure disruption [70]. By promoting metabolic reprogramming, this dynamic imbalance not only jeopardizes the structural integrity of mitochondria but also fosters the malignant development of malignancies [71,72]. Recent research has shown that TC cells exhibit notable aberrant mitochondrial dynamics, primarily characterized by enhanced fission processes and decreased fusion function, and that this imbalance is directly linked to the development of TC and the malignant phenotype [72]. For example, in ATC, a marked dysregulation of mitochondrial dynamics is observed, characterized by upregulated expression of the fission protein Drp1 and downregulated expression of the fusion protein Mfn2. This imbalance suppresses OXPHOS activity, promotes metabolic reprogramming, and ultimately drives ATC malignant progression [21,71,72], demonstrating a direct correlation between aberrant mitochondrial dynamics and the aggressive behavior of ATC.

Mitochondrial fission promotes TC invasion. It has been demonstrated that the *BRAF V600E* mutation leads to aberrant mitochondrial dynamics in TC. Specifically, the *BRAF V600E* mutation activates the *BRAF V600E*/p-ERK/p-DRP1 (Ser616), signaling cascade response, promoting mitochondrial hypersegmentation and the reprogramming of glucose metabolism, and enhancing PTC cells’ survival, growth, and proliferation [47,48]. The use of midivi-1, a Drp1 inhibitor, dramatically decreased the invasive and metastatic capacity of tumor cells in thyroid eosinophilic cell tumors, according to Ferreira-da-Silva et al. [68,73]. This study offers a new intervention technique for the targeted strategy of TC.

Mitochondrial fusion improves the prognosis of TC. Through the analysis of TCGA data, Mi-Hyeon You et al. discovered that Mfn2 expression in normal thyroid tissues was significantly higher than that in cancerous tissues. They also discovered a significant correlation between high Mfn2 expression and the characteristics of *RAS* mutations, high differentiation, and a lower rate of lymph node metastasis [46]. Subsequently, Mi-Hyeon You’s team found, through experimental analysis, that downregulation of Mfn2 expression could enhance the invasion ability of ATC cells by activating the PI3K-AKT signaling pathway, inducing EMT [46]. According to the aforementioned research, Mfn2 is a key regulatory molecule of TC malignancy. Its expression level not only acts as a biomarker for determining the differentiation status of tumors but also has the potential to be a significant therapeutic target and prognostic indicator.

### 4.3. Mitochondrial Autophagy: The Guardian of TC

Autophagy is a highly regulated process of lysosomal protein degradation that eliminates damaged or excess organelles and protein aggregates, thereby preserving intracellular homeostasis and cellular integrity. According to recent research, mitochondrial autophagy, as a selective form of autophagy, relies on both PINK1/Parkin-dependent and non-PINK1/Parkin-dependent pathways to selectively clear mitochondria that are functionally impaired [21,74,75,76]. This process exhibits a unique biphasic regulatory nature in tumor biology. Moderate mitochondrial autophagy enhances tumor cells’ ability to survive and proliferate by maintaining mitochondrial homeostasis, reducing ROS levels, and promoting OXPHOS [77,78]. In contrast, excessive mitochondrial autophagy activation may trigger “mitophagic catastrophe,” which potentiates anti-tumor immune responses through metabolic and functional modulation of immune cells [79,80,81]. Notably, systematic research on the specific regulatory mechanisms of mitochondrial autophagy and its biological effects in the development of TC is still lacking. Further investigation in this area will offer a new theoretical foundation and potential targets for intervention in the treatment of TC.

Junguee Lee and Sujin Ham’s team discovered that dysfunctional mitochondrial autophagy, caused by down-regulated Parkin protein expression, leads to an abnormal build-up of ROS in thyroid eosinophilic cell tumors. This accelerates tumor development by causing genomic instability and triggering pro-survival signaling pathways, such as NF-κB [49]. In mouse model studies, it was discovered that knocking out the mitochondrial endocytosis-associated protein—MIEAP, a key molecule involved in atypical mitochondrial autophagy—led to the development of *BRAF V600E*-driven PTC occurring more quickly. This suggests that mitochondrial autophagy is an important tumor-suppressing mechanism in TC [82]. Subsequently, research employing the mt-mKeima probe verified that curcumin induces mitochondrial autophagy, thereby dramatically reducing PTC cell proliferation. Moreover, it exhibits synergistic therapeutic potential when used with RAI [83]. All in all, the aforementioned research concluded that mitochondrial autophagy inhibition could enhance the effectiveness of RAI. Further investigation into its clinical efficacy is necessary to improve the sensitivity of patients with advanced and poorly differentiated TC to RAI therapy, ultimately boosting the patients’ clinical prognosis and survival.

### 4.4. Mitochondrial Permeability: A Switch for Programmed Cell Death in TC Cells

Emerging research has demonstrated that oxidative stress and a hypercalcemic microenvironment can dynamically regulate and control the opening status of the mitochondrial permeability transition pore (mPTP). Under physiological conditions, transient mPTP opening participates in the precise regulation of intracellular calcium ions and ROS signaling pathways. Under pathological conditions, however, sustained mPTP opening establishes a positive feedback loop via the “ROS-induced ROS release” (RIRR) mechanism, leading to amplified oxidative stress and progressive cellular damage [24,84]. On the one hand, this pathological process causes mitochondrial energy metabolism to malfunction and activates Cyclophilin D (CypD)-dependent cell death pathways [85,86,87]. On the other hand, it causes mtDNA to escape into the cytoplasm as damage-associated molecular patterns, mediated by the mPTP and mitochondrial voltage-dependent anion channels (VDAC), which are recognized by the cGAS-STING pathway. This, in turn, triggers the innate immune response and pro-inflammatory signaling cascade, which subsequently activates programmed cell death, including apoptosis and pyroptosis [88,89].

The change in mitochondrial permeability, which controls apoptosis, pyroptosis, and mtDNA-mediated inflammatory responses, is essential for the development of malignant tumors, including lung cancer and melanoma. Increased permeability of both mitochondrial membranes—inner membrane permeability and outer membrane permeability—triggers a cascade of events, including the collapse of mitochondrial membrane potential, cytochrome c release, and mtDNA leakage. These pathological changes subsequently activate caspase-3-dependent apoptotic cascades and gasdermin D-mediated pyroptosis. MtDNA that escapes into the cytosol activates the cGAS-STING pathway to initiate inflammatory responses, exhibiting dual roles in the tumor microenvironment: it can potentiate anti-tumor immunity while paradoxically fostering tumor progression under certain conditions [90,91,92,93]. According to a recent study, *BRAF V600E* in *BRAF V600E* mutant TC inhibits tumor cell death by regulating mitochondrial permeability transition through the pERK-pGSK-CypD signaling axis [50]. This discovery offers a new target for the targeted therapy of TC.

### 4.5. Disorders of Mitochondrial Metabolism

It is believed that the primary function of mitochondria, the central organelles of cellular energy metabolism, is the catabolism of glucose, lipids, and amino acids [40]. Increasing evidence suggests that metabolic reprogramming is a crucial aspect of carcinogenesis, promoting tumor growth by regulating and controlling pathways associated with cell proliferation and invasion [41,94]. More critically, metabolic reprogramming fully reflects the core position of mitochondrial metabolism in tumor biology and raises the risk of metastasis in addition to promoting primary tumor growth [95].

Glucose metabolism. TC subtypes exhibit distinct glycolytic heterogeneity, with their glycolytic activity demonstrating a nonlinear, U-shaped relationship with the degree of differentiation. In terms of molecular mechanisms, highly differentiated TC (such as PTC) partially retain the metabolic traits of normal thyroid follicular cells, mostly displaying an OXPHOS-dominated metabolic pattern, while maintaining basal levels of glycolytic activity to sustain cell proliferation. Notably, the *BRAF V600E* mutant PTC showed a comparatively suppressed glycolytic phenotype, which could be related to the quantity of sodium iodide symporter (NIS). In contrast, there is a notable reprogramming of glucose metabolism in poorly differentiated and dedifferentiated TC: To meet the energetic and biosynthetic needs of malignant tumor cell proliferation, the glycolytic pathway is activated by the upregulation of glucose transporter proteins, hexokinase 2, lactate dehydrogenase A, and pyruvate dehydrogenase kinase 1. This is accompanied by the progressive degradation of mitochondrial OXPHOS function [19,23]. These findings not only mirror the fundamental biological divergence among TC subtypes but also unveil a therapeutic window for precision metabolic targeting. For instance, *BRAF V600E* mutant PTC may benefit from a synergistic intervention that targets the glucose transporter protein-mediated glucose uptake pathway or combines with *BRAF* inhibitors. For other subtypes, however, a combination treatment regimen consisting of glucose metabolism modulators (such as 2-deoxyglucose) and differentiation inducers (such as PPARγ agonists) may improve clinical prognosis by promoting tumor cell redifferentiation and enhancing epigenetic modification. Furthermore, the pentose phosphate pathway is highly active in TC, producing ribulose-5-phosphate and NADPH, which promote tumor growth and antioxidant defense. According to Chien-Liang Liu et al., the combined inhibition of pentose phosphate pathway using oxythiamine (a transketolase inhibitor) and 6-aminonicotinamide (a G6PD inhibitor) could promote apoptosis and repress the proliferation of PTC and ATC cells by increasing ROS [96].

Lipid metabolism. A further significant characteristic of TC is abnormal lipid metabolism [54]. Through the AKT/mTOR/SREBP1c pathway and the PC-ACLY-FASN metabolic axis, pyruvate carboxylase overexpression in PTC stimulates fatty acid synthase (FASN) and acetyl-CoA carboxylase (ACC), promoting fatty acid production and enhancing PTC cells’ proliferation and invasion [52]. Furthermore, PTC enhances cellular fatty acid intake, which facilitates fatty acid entry and intracellular use primarily via key proteins such as CD36, FATP, and FABPs [53]. A significant correlation was found by Giovanna Revilla et al. between TC invasive and aberrant cholesterol metabolism. Tumor tissues exhibited typical metabolic reprogramming, characterized by the upregulation of low-density lipoprotein receptor (LDLR) expression and downregulation of 3-hydroxy-3-methylglutaryl-coenzyme A reductase and 25-hydroxycholesterol 7-alpha-hydroxylase, despite lower serum LDL-C levels in patients with high-risk PTC and ATC. This led to the aberrant accumulation of pro-carcinogenic metabolites, such as 27-hydroxycholesterol (27-HC). Furthermore, in vitro tests verified that LDL enhances the ability of ATC cells to proliferate and metastasize. The aforementioned results suggest that the cholesterol-27-HC axis relies on a distinct metabolic reprogramming mechanism to drive the malignant development of TC [97], providing a theoretical foundation for treatment approaches that target cholesterol metabolism.

Amino acid metabolism. Significant reprogramming of glutamate metabolism was observed in PTC, characterized by abnormally high expression of the glutamine transporter protein SLC1A5, glutaminase, and glutamate dehydrogenase. This metabolic adaptation offers key support for PTC cells’ biosynthetic needs [54]. In TC, Woo Young Sun et al. discovered that serine/glycine metabolic reprogramming exhibits a notable subtype specificity. In particular, PTC (particularly *BRAF* mutant) had high expression of phosphoglycerate dehydrogenase and serine hydroxymethyltransferase 1 (SHMT1), whereas FTC and MTC had low expression of both, and ATC had high expression of SHMT1. The aforementioned changes were associated with a shorter survival period [98]. Furthermore, one-carbon metabolism and serine play significant roles in ATC. Seong Eun Lee’s team discovered that high expression of SHMT2 and methylenetetrahydrofolate dehydrogenase 2 was closely linked to a poor prognosis and low grade of ATC [51]. Significantly, the capacity of ATC cells to proliferate and invade was successfully suppressed by intervention with SHMT2-specific inhibitors, indicating that SHMT2 may be a potential therapeutic target for ATC [51].

All things considered, the aforementioned research shows that mitochondrial metabolic dysfunction is strongly linked to the malignant development of TC. Further research into metabolic features of TC subtypes is necessary to help develop the exact targeted treatment of TC.

### 4.6. Mitochondrial Dysfunction and the TC Tumor Microenvironment (TME)

The TME, a complex environment that supports the survival and growth of tumor cells, comprises cancer cells, non-cancerous cells, and non-cellular components that all cooperate to contribute to tumor development [99]. Emerging studies have revealed a complex regulatory network connecting mitochondrial dysfunction in tumor cells to immune reshaping within the TME. Notably, tumor cells with mitochondrial dysfunction can transfer mutated mtDNA to neighboring immune cells, thereby compromising T-cell functionality and facilitating immune evasion and tumor progression [100]. Through metabolic remodeling, immune cells affect tumor cell metabolic dependency (such as lactate use), which accelerates tumor development [101], providing important targets for the development of novel combination therapy strategies.

According to recent research, cancer cells can utilize either direct or indirect mitochondrial transfer mechanisms to transmit mutant mtDNA to tumor-infiltrating lymphocytes, thereby controlling tumor immune evasion through the transfer of mtDNA. Mutant mtDNA originating from cancer cells causes T cells to undergo senescence and mitochondrial dysfunction, along with diminished cellular activity, metabolic abnormalities, and impairments in effector function and memory formation. In turn, this degrades anti-tumor immunity both in vivo and in vitro, and lessens the efficacy of PD-1 blocking treatment [100]. This discovery not only clarifies a new immune escape route for cancer but also offers a theoretical foundation for developing combination immunotherapy approaches that target mitochondrial metastases.

Casimiro et al. discovered that, in breast cancer, the aberrant high expression of the mitochondrial fission-associated protein Mff causes metabolic remodeling of cancer-associated fibroblasts to an aerobic glycolytic phenotype, resulting in the accumulation of lactic acid. The breast cancer cells then absorb this lactic acid to enter the tricarboxylic acid cycle and undergo OXPHOS, or “Two-Compartment Tumor Metabolism,” which effectively promotes early tumor growth [101]. The idea that “tumor is a disease of TME” is supported by these findings, which provide important insights into the causal link between abnormal mitochondrial dynamics and metabolic abnormalities in TME.

The function of mitochondrial autophagy in controlling and regulating the tumor immunological milieu has seen significant developments. Studies have demonstrated that mitochondrial autophagy has two functions in immune regulation [102]: when in an immune activation condition, it provides energy and improves the activity of immune cells like B cells, NK cells, CD8+ T cells, and macrophages; when the tumor immunosuppressive microenvironment is present, mitochondrial autophagy helps tumor cells avoid immune surveillance. Not only does mitochondrial autophagy regulate tumor-associated macrophages (TAMs)’s metabolism toward M2 polarization, thereby promoting immunosuppression, but it also inhibits metabolic activities in effector T cells and dendritic cells, thereby reducing their ability to fight tumors and fostering immunological tolerance [103]. Targeting the control of mitochondrial autophagy has, therefore, become a viable approach for immunotherapy.

In summary, the regulatory role of mitochondrial function in the TME has been widely recognized. Although the precise mechanisms underlying its involvement in TC TME regulation remain incompletely understood, further elucidation of these molecular pathways may contribute to enhancing the efficacy of immunotherapy for TC.

### 4.7. Mitochondrial Dysfunction and TC Stem Cells (TCSCs)

CSCs are a subset of cancer cells that possess the capacity for self-renewal, multilineage differentiation, resistance to therapy, and the ability to migrate and metastasize. Their plasticity and interplay with mitochondrial dysfunction drive tumor initiation, progression, and drug resistance [104]. Through distinct mitochondrial dysfunction (metabolic reprogramming, elevated mitochondrial membrane potential, and low-energy production), PTCSCs have been shown to retain their stemness and develop treatment resistance [105]. In the ATC investigation, Mika Shimamura et al. subsequently demonstrated that CSCs maintain low ROS levels by reducing mitochondrial OXPHOS activity, which enhances their stemness and plasticity [106]. This discovery further confirms that mitochondrial dysfunction of ATCSCs promotes the development and occurrence of cancer. Recent data suggest that the synergistic interaction between mitochondrial autophagy and mitochondrial biogenesis regulates the fate of CSCs. Mitochondrial biogenesis improves energy metabolism, increases the number and quality of mitochondria, and mitochondrial autophagy removes damaged mitochondria while maintaining ROS homeostasis. These processes support CSC survival, metastasis, and treatment resistance. However, the excessive activation of both triggers CSCs’ death in reverse. HIF-1, AMPK, MYC, and other mechanisms precisely control this balance, allowing CSCs to resist mitochondrial damage while retaining their stemness, plasticity, and treatment resistance [107]. Low mtDNA-CN is characteristic of CSCs, which contributes to their resistance to treatment and preserves their stem cell-like characteristics. Research has found that lowering the mtDNA-CN activates the calcineurin-dependent mitochondrial retrograde signaling pathway, inducing EMT, gaining a migratory/invasive phenotype, and promoting the production of self-renewing CSCs. Restoring the mtDNA content, however, undoes the above changes [108,109]. As a result, CSCs resistant to traditional treatments can be successfully cleaned up by inhibitors of mitochondrial retrograde signaling, which also inhibits tumor growth and reduces harm to healthy tissues. However, the above mechanism has not been thoroughly elucidated in TCSCs and warrants further investigation to develop novel therapeutic strategies for TC.

### 4.8. Mitochondrial Dysfunction and TC Treatment Resistance

For patients with iodine-refractory, locally advanced TC, targeted therapy has advanced quickly in recent years. The intricacy of medication resistance mechanisms, however, has emerged as a significant problem hindering improvement in patient prognosis. The two types of drug resistance in TC are primary resistance and acquired resistance. Primary resistance is primarily due to the tumor cells’ lack of reliance on the therapeutic target, or the presence of alternative targets, resulting in the drug failing during initial treatment. The capacity of cancer cells to progressively acquire drug resistance by adaptive changes (such as genetic mutations, signaling pathway activation) under the stress of treatment is known as acquired resistance [110]. Key mechanisms of kinase inhibitor resistance include MAPK/PI3K pathway reactivation, ABC transporter upregulation, EMT induction, CSCs enrichment, suppressed apoptosis, altered autophagy, metabolic reprogramming toward aerobic glycolysis, and TME-mediated drug tolerance [111]. On the other hand, the primary cause of RAI resistance is the aberrant expression or localization of NIS, which is brought on by the regulation of non-coding mRNAs, alterations in signaling pathways like MPKA/PI3K (which are caused by gene mutations like *BRAF V600E*), and inhibition of the thyroid-stimulating hormone receptor signaling pathway [112]. Drug resistance in TC is significantly influenced by mitochondrial function and the metabolic pathways linked to it, according to existing research [113,114].

Mitochondrial metabolism. Significant changes in TC cells’ metabolic properties can be seen, such as increased synthesis of fatty acids, hyperglutamine metabolism, and enhanced glycolysis. Numerous investigations have confirmed the strong correlation between these metabolic changes and medication resistance [111]. For instance, Vlad C. Sandulache et al. experimentally discovered that 2-Deoxy-D-glucose (2-DG)-induced glycolysis inhibition enhanced the susceptibility of ATC cells to radiation and cisplatin [115]. Subsequently, it was discovered that 2-DG could inhibit increased glycolysis in *BRAF V600E*-mutated PTC cells, thereby increasing the drug sensitivity of PTC cells to sorafenib [116]. This further demonstrated the tight relationship between primary drug resistance in TC and mitochondrial metabolic reprogramming. Similar findings were reported by Veronica Valvo et al., who discovered that *BRAF V600E* PTC downregulated ACC2 expression, and that treatment with vemurafenib increased ACC2 expression. However, ACC2 knockdown reversed this effect, increased vemurafenib resistance, and promoted tumor growth [117], indicating that targeting ACC2 may be a potential strategy for overcoming resistance to *BRAF V600E* inhibitors.

Mitochondrial autophagy. It has been discovered that mitochondrial autophagy preserves homeostasis by eliminating damaged mitochondria, increases the supply of OXPHOS, and lowers ROS levels, which keeps drug-resistant persisters alive and inhibits apoptosis. It also encourages the formation of acquired resistance to cisplatin, adriamycin, and other cancer chemotherapeutic drugs [79,80]. As previously indicated, the induction of mitochondrial autophagy through curcumin promotes ROS bursts and cytotoxicity, thereby dramatically increasing the susceptibility of PTC cells to RAI therapy [83]. Mitochondrial autophagy is closely linked to acquired drug resistance and may be a potential therapeutic target, as indicated by the aforementioned research.

In addition, it has also been reported that OPA1-mediated suppression of cytochrome c release is linked to venetoclax resistance in acute myeloid leukemia (AML), which is an example of acquired drug resistance [118]. In summary, TC resistance is strongly linked to mitochondrial metabolism, autophagy, and mitochondrial dynamics, affecting both acquired and primary resistance, with a preference for acquired resistance. Given the complexity of mitochondrial function, future studies should further elucidate its role in TC drug resistance to enhance the clinical efficacy of targeted therapies (such as *BRAF* inhibitors).

## 5. Mitochondrial Information Processing Networks: Bridging Mitochondrial Dysfunction and TC Progression

Although mitochondria have traditionally been believed to be primarily responsible for energy and metabolic processing, as research has advanced, it has become increasingly evident that they play a crucial role in the field of information processing. As a representation of endosymbiotic integration in TC cells, mitochondria rely on the mitochondrial information processing system (MIPS) to integrate various pieces of information, including ions, proteins, nutrients, and energy status. Through the endoplasmic reticulum (ER) and the mitochondrial physical communication platform, which is constructed by mitochondria-associated membranes (MAMs), mitochondria also accomplish trans-organelle information interactions [119]. Decoding the MIPS-integrated information can activate particular genetic programs that control metabolic change, which in turn propels TC cell survival, proliferation, metastasis, and drug resistance processes, all of which work together to further TC cells’ pathological process (Figure 3) [120].

### 5.1. Calcium

Calcium ions are released through the IP3R/RyR channels of the ER, transferred to mitochondria via the GRP75-VDAC complex in MAMs, and then enter the mitochondrial matrix via the mitochondrial calcium uniporter [121,122,123]. Furthermore, TC has been shown to upregulate the oncogenic channel transient receptor potential vanilloid-6 (TRPV6), which mediates extracellular calcium inward flow and indirectly influences mitochondrial calcium homeostasis [124]. In TC, through sodium–calcium exchangers, calcium ions from the mitochondria are released into the cytoplasm, activating NF-κB, calcium/calmodulin-dependent kinase II (CaMKII), and other crucial calcium-dependent factors (such as calcium-dependent serine–threonine phosphatase). This ultimately promotes TC cell survival, proliferation, migration, and immune evasion, while inhibiting apoptosis [125,126]. Building on this, a phase I clinical trial in patients with advanced epithelial-derived tumors (NCT01578564; https://clinicaltrials.gov/study/NCT01578564, accessed on 25 August 2025) demonstrated that inhibition of TRPV6 calcium channels using soricimed biopharma’s compound (SOR-C13) resulted in disease stabilization in 54.5% of patients, with a duration ranging from 2.8 to 12.5 months, while maintaining favorable drug tolerability [127].

The diverse cellular origins of TC cells lead to varied patterns of calcium signaling regulation. CaMKII promotes tumor growth in follicular cell-derived PTC by activating the ERK pathway signaling in response to *RAS* or *RET/PTC* oncogenes [128]. Because MTC cells have special calcium-sensing and calcitonin-secreting abilities, prolonged activation of CaMKII caused by RET mutations in c cell-derived MTC not only maintains proliferative signals but also disturbs the delicately balanced regulation of hormone production. Notably, the clinical severity of MTC is negatively correlated with the expression level of hCaKIINα, an endogenous inhibitor of CaMKII [125], suggesting its potential as a biomarker and therapeutic target for MTC. The aforementioned results demonstrate the tissue-specific origin of calcium signaling pathways, offering a vital basis for the targeted treatment of various TC subtypes.

### 5.2. ROS

The ER-associated NADPH oxidase 4 and endoplasmic reticulum oxidoreductase 1α in TC produce ROS, which can diffuse into the mitochondria. The IP3R-GRP75-VDAC/Sig-1R complex controls the transfer of calcium ions from the ER to the mitochondria. Electron transport chain (ETC) dysfunction and electron leakage in ETC complexes I and III work in concert to create a positive feedback loop known as RIRR. This further increases the generation of ROS, allowing them to diffuse into the nucleus as hydrogen peroxide [129,130]. Long-term chronic oxidative stress adaptation promotes the production of inflammation, TC cell proliferation, and immune escape by activating pro-TC cell survival signaling pathways, such as PI3K/AKT and NF-κB. At the same time, it upregulates antioxidant defenses like catalase and superoxide dismutase, which reduce ROS toxicity, help tumors adapt to microenvironmental stresses, and contribute to the formation of drug resistance [131]. For instance, Jérôme Alexandre discovered that, in breast cancer, the activation of plasma membrane NADPH oxidase increases the ROS concentration, which is correlated with paclitaxel resistance [132].

### 5.3. Product of Metabolism

According to new research, tumor metabolites also influence epigenetics, establishing a metabolic–epigenetic regulatory axis that promotes the occurrence and development of tumors [133]. According to Xumeng Wang et al., for instance, ATC with the *BRAF V600E* mutation increases glycolysis through the Warburg effect, resulting in a significant production of lactic acid. This substance acts as a precursor that not only satisfies the energy requirements of tumor cells by causing histone lysine lactylation (such as H4 K12 La), but also activates the expression of several genes necessary for ATC proliferation, thereby promoting ATC proliferation [134]. The aforementioned results suggest that the development of medications that block lactate generation and reverse histone lactylation, in conjunction with *BRAF V600E* inhibitors, will likely inhibit the progression of ATC. Concurrently, targeting the bidirectional signaling axis connecting mitochondrial metabolism and epigenetics, in combination with existing *BRAF* inhibitors, presents a novel therapeutic strategy for treating patients with TC.

In conclusion, in TC, metabolic reprogramming (such as the Warburg effect) jointly promotes cancer through lactate-mediated histone lactonylation with epigenetic modifications, while calcium ions and ROS drive the progression of malignant tumors through a bidirectional ER–mitochondria–nucleus conductance network. These findings suggest that targeting TRPV6 (such as SOR-C13) or combining metabolic–epigenetic interventions (such as lactate inhibitors and *BRAF* inhibitors) may represent novel therapeutic strategies for addressing this condition. However, their clinical translational potential requires further validation.

## 6. Key Regulatory Pathways of Abnormal Mitochondrial Functions in TC

### 6.1. AMPK-PGC-1α-NRF1

AMPK, functioning as a cellular energy sensor, is activated through the detection of alterations in AMP/ATP or ADP/ATP ratios [135]. Activated AMPK phosphorylates peroxisome proliferator-activated receptor gamma coactivator 1α (PGC-1α), which in turn influences nuclear respiratory factor 1 (NRF1) transcriptional activity, contributes to the control of mitochondrial biosynthesis, and keeps OXPHOS function [136,137,138]. The aforementioned pathways are typically inhibited in TC, resulting in the downregulation of PGC-1 expression, decreased mitochondrial biosynthesis, metabolic reprogramming towards glycolysis, a decreased ability to clean up ROS, and increased genomic instability. All of these factors contribute to the development of TC [139]. Thus, a potential target for TC treatment could be the AMPK-PGC-1α-NRF1 pathway **(**Figure 4).

### 6.2. PI3K-AKT-mTOR

The PI3K-AKT-mTOR pathway is a key regulatory hub for cell growth, metabolism, and survival. Through complex signaling cascades, it contributes to protein synthesis, energy consumption, apoptosis suppression, and cell proliferation. Cancer and abnormal activation of this pathway are closely associated [140]. In TC, PI3K-AKT-mTOR pathway aberrant activation stimulates mitochondrial biogenesis and metabolic reprogramming via a mTORC1-dependent mechanism, which supplies the energy and material basis for tumor cells’ proliferation and drives the malignant progression of TC [141] (Figure 4).

### 6.3. Wnt/β-Catenin

The Wnt/β-catenin pathway is a highly conserved cellular signaling pathway that controls important biological processes, including cancer occurrence, stem cell maintenance, differentiation, cell proliferation, and embryonic development [142]. The Wnt/β-catenin pathway is abnormally activated in TC, which accelerates cell cycle progression and drives metabolic reprogramming by transcriptionally upregulating key target genes, such as Cyclin D1 and c-Myc, ultimately promoting tumor cell proliferation [143,144]. In the meantime, this pathway induces EMT, resulting in a loss of cell polarity and a reduction in cell junctions, which significantly enhances the tumor cells’ ability to invade and migrate [145] (Figure 4).

## 7. The Potential of Mitochondrial Dysfunction as a Therapeutic Target for the Treatment of TC

Significant therapeutic advancements have been made in the management of TC in recent years. The integration of surgery, targeted medicine, and RAI treatment has markedly improved the prognosis for a substantial proportion of TC patients. However, the prognosis is still dismal for patients with ATC, MTC, locally advanced PTC, and some iodine-refractory and drug-resistant PTC. Research has shown that TC pathogenesis and drug resistance are closely linked to mitochondrial dysfunction. To address these unmet clinical needs, new therapeutic strategies that target mitochondrial function may offer an innovative direction to overcoming the current therapeutic bottlenecks, which have significant clinical value. The following text systematically summarizes various intervention strategies targeting mitochondrial dysfunction and examines their clinical value when combined with existing targeted therapies, RAI therapy, and mitochondrial transplantation in TC (Figure 5; Table 2 and Table 3).

### 7.1. Targeting Mitochondrial Metabolism

Fasting or fasting mimicking diet (FMD). It has been discovered that fasting or FMD can encourage a change in the metabolism of tumor cells from aerobic glycolysis to OXPHOS. This change increases the production of ROS and enhances the curative effect of chemotherapeutic agents, showing promise as an adjuvant treatment for malignant tumors, such as lung and breast cancer [155]. In an in vitro and in vivo model, Xiaoping Zhang’s group found that FMD therapy can suppress PTC cell growth and proliferation [156]. The potential of FMD for clinical use was further confirmed by a phase II clinical trial that included 131 patients with breast cancer (NCT 02126449; https://clinicaltrials.gov/study/NCT02126449, accessed on 25 August 2025). The trial demonstrated that FMD significantly increased tumor response rates (OR 3.168, *p* = 0.039), enhanced pathological response (OR 4.109, *p* = 0.016), and reduced chemotherapy-related adverse events (AE) [157].

Tigecycline. Tigecycline has been shown to have anticancer potential by inhibiting OXPHOS, causing oxidative stress, and encouraging mitochondrial biogenesis, thus inhibiting the progression of malignant tumors such as AML and liver cancer [158]. Studies about tigecycline in TC are currently in the preclinical stage. Reed GA et al. discovered that the maximum tolerated dose of tigecycline in a phase I clinical trial of adult patients with relapsed and refractory AML (NCT 01332786; https://clinicaltrials.gov/study/NCT01332786, accessed on 25 August 2025) was 300 mg/day in 27 patients treated with tigecycline. This medication had good safety but did not demonstrate efficacy due to its short half-life, resulting in insufficient blood concentrations. To improve efficacy, better dosing regimens (such as continuous infusion) are required in the future [150].

### 7.2. Targeting Mitochondrial Dynamics

Mdivi-1. Mdivi-1 was once thought to be an inhibitor of the protein Drp-1, which is involved in mitochondrial fission. However, additional investigation showed that Mdivi-1 can inhibit OXPHOS function by reversibly inhibiting Complex I through a mechanism that is independent of Drp-1 or mitochondrial fusion. It eventually showed promise as an anticancer agent in cellular studies by inhibiting the growth of cancerous tumor cells, including those of the lung and colon [159,160]. In the PTC and ATC cell trial, Lin Zhang’s group found that the application of mdivi-1 enhanced apoptosis and significantly reduced the proliferation, invasion, and metastatic capacity of PTC and ATC cells [146].

Ruxolitinib. The JAK1/2 inhibitor rufolitinib has been shown to inhibit the growth of gastric cancer and other malignant tumors by blocking JAK/STAT3 signaling, disrupting the equilibrium of mitochondrial dynamics, and inducing oxidative stress [161]. This suggests that rufolitinib has the potential to treat tumors. In an in vitro and in vivo model of the thyroid gland, Ya-Wen Guo et al. discovered that the use of rufolitinib greatly inhibited the proliferation and encouraged the apoptosis and pyroptosis of PTC and ATC cells. Ruxolitinib markedly and dose-dependently inhibits tumor growth in a mouse model of ATC xenograft tumors without showing signs of drug toxicity [162]. In Phase II clinical trials for 44 patients with chronic myeloid leukemia and atypical chronic myeloid leukemia (NCT 02092324; https://clinicaltrials.gov/study/NCT02092324, accessed on 25 August 2025), ≥Grade 3 anemia and thrombocytopenia were observed in 34% and 14% of patients, respectively, following treatment with ruxolitinib. No serious AE attributable to ruxolitinib were observed [151]. The aforementioned results demonstrate ruxolitinib’s enormous potential for tumor treatment. Even though the TC study is still at the preclinical level, it offers basic guidelines for its possible effectiveness in treating mitochondrial TC.

### 7.3. Targeting Mitochondrial Oxidative Stress

Antioxidants. As natural antioxidants, vitamin C and vitamin K have been shown to inhibit tumor cell growth and metastasis in melanoma, hepatocellular cancer, and other malignant tumors by increasing ROS production, thereby causing oxidative stress, and enhancing the tumor infiltration capabilities of T and NK cells. Additionally, when vitamin C is combined with vitamin K3, it can increase tumor cell sensitivity to chemotherapeutic drugs [163], indicating that natural antioxidants have potential anti-tumor effects [164]. Through experimental research, Xi Su’s team demonstrated that vitamin C has a negligible effect on normal cells but significantly suppresses the proliferation of PTC, FTC, and ATC cells, causing death in a time- and dose-dependent manner by inducing a substantial increase in ROS. High-dose vitamin C (4 g/kg) therapy significantly decreased the weight and volume of ATC-derived xenografts in an in vivo mouse model, substantially inhibiting tumor growth [165]. A phase III clinical trial in 442 patients with unresectable, treatment-naïve metastatic colorectal cancer (NCT 03146962; https://clinicaltrials.gov/study/NCT03146962, accessed on 25 August 2025) demonstrated that, in RAS-mutated patients, the combination of high-dose vitamin C with standard chemotherapy significantly prolonged median progression-free survival to 9.2 months compared to chemotherapy alone. These findings suggest that vitamin C selectively targets RAS-mutated colorectal cancer cells while maintaining a favorable safety profile when combined with chemotherapy [153].

Alantolactone (ALT). It has been discovered that ALT has anti-tumor effects, primarily by utilizing the ROS-dependent caspase pathway to induce apoptosis and the GSDME-dependent pyroptosis pathway in a range of malignant tumors, including lung cancer and ATC [166]. Yiqun Hu et al. discovered that ATL can inhibit the proliferation of ATC cells. Moreover, ATL exhibits remarkable safety, activating proteins linked to apoptosis and inhibiting the growth of subcutaneous xenograft models of ATC [148].

Shikonin. By causing ROS-mediated DNA damage and activating the p53 signaling pathway, Qi Yang et al. discovered that shikonin induced apoptosis in TC cells and downregulated matrix metalloproteinase expression, thereby inhibiting the EMT process and significantly decreasing the TCs’ capacity to metastasize, thereby elaborating on its potential anti-tumor properties [149]. Yang et al. further discovered, through cell and animal experiments, that shikonin can induce cell cycle arrest and apoptosis, as well as inhibit the proliferation, migration, and invasion of PTC, FTC, and ATC cells. Notably, shikonin significantly reduces the proliferation of TC cells in mutant p53TC cells and primary cancer cells. Shikonin demonstrated strong anticancer effectiveness and a safe profile in the FTC nude mice xenograft model by drastically inhibiting tumor growth without causing liver damage [149].

Photodynamic (PDT). Evidence shows that PDT can cause tumor cells to undergo apoptosis and necrosis by generating ROS through photosensitizers activated by light. It can also destroy blood vessels and trigger immune responses, which can have anti-tumor effects on malignant tumors like basal cell carcinoma and laryngeal carcinoma. These findings reveal the potential ability of PDT in tumor treatment [167]. In cellular and animal experiments, Hyejin Kim et al. discovered that hypericin-PDT efficiently suppresses the growth of ATC cells and induces apoptosis. Additionally, the ATC tumors in mice were successfully eliminated by hypericin-PDT. Based on this, hypericin demonstrates potential application value as a photosensitizer in the treatment of ATC [168]. Ellen J. Kim’s team discovered that patients treated with three cycles of hypericin-PDT had a lesion response rate of up to 49% in a phase III clinical trial of 169 patients with early-stage cutaneous T-cell lymphoma (mycosis fungoides) (NCT 02448381; https://clinicaltrials.gov/study/NCT02448381, accessed on 25 August 2025). The most frequent treatment-related AE were mild localized cutaneous (17.3%) and dressing-site reactions (6.9%), and there were no serious drug-related AE [169]. The aforementioned results further support the enormous potential of PDT in anti-tumor treatment.

### 7.4. Targeting Mitochondrial Membrane Potential

Myricetin. It was discovered that myricetin disrupted the mitochondrial membrane potential and induced apoptosis through the mitochondrial pathway and other mechanisms, leading to increased apoptosis and decreased proliferation of malignant tumor cells, such as TC and pancreatic cancer [170], which demonstrates its antitumor potential. Tae Kwun Ha and colleagues found that myricetin exhibits dose-dependent cytotoxicity against PTC cells through experimental studies, reducing cell viability, inducing apoptosis, and inhibiting proliferation [147]. However, the application of myricetin in cancer therapy still lacks further validation through clinical trials, necessitating in-depth investigations into its antitumor mechanisms and therapeutic potential in future research.

Mitotane. Research has revealed that mitotane promotes apoptosis in TC cells by disrupting mitochondrial membrane potential, triggering a surge in ROS and cytochrome c release, which subsequently activates the caspase cascade [93]. In cellular experiments, Athanasios Bikas et al. discovered that mitotane inhibited proliferation and promoted apoptosis in PTC, FTC, MTC, and ATC cells. Notably, MTC cells were more susceptible to mitotane, which may be because MTC cells express more ATP5B [93]. Clinical consensus has been reached regarding the adjuvant treatment of adrenocortical tumors with mitotane [171], further confirming its efficacy and safety in tumor therapy.

Niclosamide. Research has demonstrated that niclosamide exerts potent anti-tumor effects by inducing mitochondrial uncoupling, which leads to membrane potential depolarization and ROS generation, ultimately triggering apoptosis in malignant cells, including leukemia and TC, and significantly suppresses tumor cell proliferation and metastatic potential [172]. Notably, niclosamide exhibits chemosensitizing properties when combined with conventional chemotherapeutic agents (such as cisplatin), enhancing their therapeutic efficacy through synergistic mechanisms [173]. Through cellular research, Kai Yu’s group discovered that niclosamide induces apoptosis and inhibits the migration, invasion, and proliferation of PTC and ATC cells, possessing potential as a treatment for TC [172]. In a phase Ib clinical trial in nine prostate cancer patients (NCT 02807805; https://clinicaltrials.gov/study/NCT02807805, accessed on 25 August 2025), the abiraterone, prednisone, and niclosamide combination induced ≥ 50% PSA reductions in 55.6% of treated patients (two complete responses), with a manageable toxicity profile supporting its further investigation for metastatic castration-resistant prostate cancer [154]. Additionally, niclosamide is being used in a Phase II trial to treat systemic or heterochronic metastases in progressing colorectal cancer (NCT 02519582; https://clinicaltrials.gov/study/NCT02519582, accessed on 4 July 2025) [174].

### 7.5. Targeting Mitochondrial Autophagy

Urolithin A (UA). It was discovered that UA may have antitumor effects on breast cancer, pancreatic cancer, and other malignant tumors by promoting memory T cell proliferation, suppressing the detrimental inflammatory response of TAMs, and enhancing mitochondrial autophagy [175,176]. Currently, there is a notable lack of clinical trial data validating the antitumor therapeutic potential of UA, necessitating further rigorous research.

Curcumin. Extracted from turmeric, curcumin is a polyphenolic molecule with a variety of pharmacological properties, such as neuroprotective, anti-inflammatory, and anti-anxiety actions. As studies continue, curcumin shows therapeutic promise in the treatment of various types of cancer, including colorectal and lung cancer [177]. Curcumin has been shown, in TC-related cellular experiments, to target mitochondria in TC cells, inducing ROS bursts and mitochondrial autophagy. This induces apoptosis and inhibits the proliferation of PTC, FTC, and ATC cells. Furthermore, curcumin exhibits synergistic anticancer effects by increasing sensitivity to RAI treatment [83].

### 7.6. Targeting Mitochondrial Calcium Homeostasis

Capsaicin (CAP). CAP promotes tumor cell apoptosis in malignant tumors, such as ATC and small cell lung carcinoma, by activating transient receptor potential vanilloid type 1 channels to induce calcium influx, leading to mitochondrial calcium overload and dysfunction [178]. Shichen Xu et al. investigated the effects of capsaicin on ATC, FTC, and PTC cells. Their results showed that capsaicin strongly suppresses ATC cell viability and promotes apoptosis in ATC cells, demonstrating a higher cytotoxic effect against ATC cells than PTC and FTC cells [179]. This implies that using capsaicin can be a potential treatment for ATC. CAP’s recent clinical trials, however, have concentrated on pain treatment (NCT 01533428; https://clinicaltrials.gov/study/NCT01533428, accessed on 25 August 2025) and have shown good safety with no notable AE [152].

### 7.7. Mitochondrial Transplantation

Mitochondrial transplantation is an emerging therapeutic modality that introduces functional mitochondria derived from healthy donor cells or synthetically reprogrammed through bioengineering techniques into recipient cells and tissues exhibiting mitochondrial dysfunction. Mitochondrial transplantation primarily utilizes autologous or heterologous sources, with delivery methods including direct injection, nanocarrier systems, and cell-mediated transfer, offering promising treatment prospects for mitochondrial-related disorders [180]. As studies progressed, it was discovered that mitochondrial transplants possess anticancer properties and are a potential treatment for tumors. By reversing the metabolic reprogramming of cancer cells, encouraging the accumulation of ROS, enhancing immune recognition, and re-establishing apoptotic pathways (such as activation of cystatin cascade reactions), transplanted mitochondria can have an anticancer effect on several malignancies, including hepatocellular carcinoma, melanoma, and breast cancer [181]. Jui-Chih Chang et al. discovered that the viability of breast cancer cells was compromised by mitochondrial transplantation using simple co-culture and Pep-1-mediated mitochondrial delivery. This was primarily seen as increased chemotherapy sensitivity, induction of apoptosis, and inhibition of breast cancer cells’ growth [182]. By decreasing aerobic glycolysis and endogenous ROS levels in hepatocellular carcinoma cells, healthy mitochondrial transplantation blocks the cell cycle. It reduces energy generation, inducing apoptosis in hepatocellular carcinoma cells and inhibiting their proliferation [183]. Following that, using in vitro and in vivo models of prostate and ovarian cancer, Aybuke Celik’s group discovered that mitochondrial transplantation dramatically reduced the tumor cells’ capacity for migration but had no discernible impact on their ability to proliferate. Low-dose chemotherapy combined with mitochondrial transplantation enhanced the susceptibility of tumor cells to chemotherapy in a manner equivalent to that of high-dose chemotherapy alone [184]. The results mentioned above suggest that mitochondrial transplantation may become a novel strategy for treating cancer. Even though there are currently no pertinent clinical trials available in TC, its potential to restore mitochondrial function, induce programmed cell death, and sensitize chemotherapy offers a promising prospect for TC treatment research. It is worthwhile to investigate its potential synergistic effect with current therapies in the future. However, in therapeutic applications, mitochondrial transplantation still faces several obstacles to overcome, including mitochondrial storage, mitochondrial viability, target delivery, and immunological rejection. First, to preserve the stability and bioenergetic function of mitochondria, techniques for mitochondrial cryopreservation must be developed. The effect of mitochondrial transplantation has also been found to exhibit time-dependent differences, with some studies confirming long-term functional recovery, while short-term bioenergetic enhancement may gradually fade [185]. Future systematic optimization of the ideal therapeutic dose is still necessary. Similarly, the development of tissue-specific mitochondrial delivery systems is crucial. This involves optimizing targeting modifications by characterizing mitochondria-specific surface recognition molecules to enhance targeting precision, improve uptake efficiency, and reduce off-target effects [186]. Lastly, some research has shown that autologous mitochondrial transplantation offers notable immunosafety advantages in response to immunological rejection [187]. Likewise, compared to isolated mitochondria, mitochondria delivered via extracellular vesicles were less immunogenic [188].

### 7.8. Combination Therapy

Myricetin and RAI therapy. RAI therapy’s rationale as an adjuvant treatment for TC stems from the fact that cancer cells’ NIS can absorb RAI [189,190]. According to Shabnam Heydarzadeh et al., myricetin significantly increased the expression of the NIS gene in ATC cells, promoting iodine absorption capacity and enhancing the efficacy of RAI treatment [191]. The aforementioned findings offer a new therapeutic approach to increase the sensitivity of RAI treatment and a new treatment strategy for RAI-refractory TC patients; nevertheless, this approach is still in the basic research stage, and rigorous clinical trials are required to confirm its clinical efficacy and safety.

Vitamin C and vemurafenib (*BRAF V600E* inhibitor). Vemurafenib, a *BRAF V600E* inhibitor, was found to be frequently resistant to feedback stimulation of the PI3K/AKT and MAPK signaling pathways when used to treat TC [192,193]. By preventing the feedback stimulation of the aforementioned signaling cascade and overcoming drug resistance, vitamin C was found by Xi Su et al. to significantly enhance the antiproliferative capacity of vemurafenib in *BRAF V600E* mutant TC cells [194]. These findings suggest that the combined use of vitamin C and vemurafenib may serve as an effective approach to overcome targeted drug resistance, offering a novel therapeutic strategy for improving prognosis in *BRAF V600E*-mutant TC patients.

DG, Adriamycin, and Sorafenib. Ling Gu and colleagues found that 2-DG, a synthetic glucose analog, selectively kills acute lymphoblastic leukemia cells. Its low dosage (250 mg/kg) shows promise as a low-toxicity anticancer medication, according to clinical research [195]. Although adriamycin (DOX), a traditional chemotherapeutic drug of the anthracycline class, can be used to treat TC, its clinical application is restricted due to its high cardiotoxicity [196]. A study of the targeted *BRAF* inhibitor sorafenib reveals toxicity consistent with its known safety profile [197]. Cellular experiments conducted by Shuo-Yu Wang’s team demonstrated that combining 2-DG with either DOX or sorafenib significantly increases the sensitivity of PTC cells to both medications, thereby enhancing their anticancer effects. At the same time, the low-toxicity 2-DG lowers the half-maximal inhibitory concentration of sorafenib and DOX, which lowers drug toxicity, and which, in turn, reduces AE and has clinical translational potential [116].

## 8. Conclusions and Perspectives

In conclusion, various types of mitochondrial dysfunction are intimately linked to the development of TC, treatment resistance, and immune microenvironment reshaping. Notably, TC cells with various cellular origins display unique biological traits related to their mitochondria: Because of their neuroendocrine origin, mitochondria in C-cell-derived MTC are essential for calcium buffering and hormone secretion regulation, while mitochondria in follicular cell-derived TC (such as PTC and FTC) usually retain partial OXPHOS, supplying energy and material preparation. In these cells, aberrant RET signaling pathways often interact with mitochondrial dysfunction to impact the development and survival of MTC cells. This type-specific mitochondrial dysfunction promotes TC progression, therapy resistance, and immune microenvironment reshaping by inducing key pathological processes such as chronic inflammation and oxidative stress, forming a synergistic vicious cycle that plays a central regulatory role in TC development. Despite tremendous progress in TC treatment, there are still no effective clinical interventions for specific pathological subtypes, such as ATC, MTC, locally advanced PTC, and some targeted therapy-resistant PTC. Accordingly, this paper reviews the molecular mechanisms of mitochondrial dysfunction for TC progression. New therapeutic approaches based on mitochondrial targeting are being examined, including targeted medications to reverse mitochondrial dysfunction and inhibit tumor growth, as well as to induce apoptosis, dietary interventions to maintain metabolic homeostasis, and the therapeutic approach of mitochondrial transplantation. New theoretical underpinnings and practical directions for creating more effective TC treatment plans are provided by incorporating these research developments.

In the realm of TC therapy, mitochondria-targeted therapeutic approaches have emerged as a significant breakthrough point. First is the development of medication that targets the mitochondria. Numerous preclinical studies have demonstrated that agents targeting the mitochondria, including chemical agents such as tigecycline, small molecule inhibitors like mdivi-1, natural active ingredients like myricetin, and the calcium inhibitors mentioned earlier, can exhibit antitumor effects by suppressing proliferation or inducing apoptosis [125,146,147,158,191,198], highlighting their significant potential for antitumor therapy. These mitochondria-targeted medications still face numerous obstacles in the clinical translation process, including the optimization of delivery systems, controlling off-target effects, tissue-specific inhibition, and evaluating long-term safety. First, nanodelivery systems. Wenjing Wang et al. studied gold-silver@polydopamine (Au-Ag@PDA) nanoparticles with mitochondria-targeting characteristics, which offer a nano-co-targeted medication therapy strategy for the treatment of iodine-refractory PTC and locally late stage PTC [199]. However, the aforementioned studies did not further evaluate the toxicity of Au-Ag@PDA. Studies on rectal cancer have discovered that Au-Ag@PDA nanoparticles still cause diffusion harm to the surrounding tissues, even though they improve biocompatibility and reduce silver ion toxicity through PDA coating. Future research should employ strategies such as conjugating cancer-specific biomarkers or utilizing cell-membrane-based biomimetic coatings to enhance active tumor targeting and prolong circulation time, thereby reducing side effects and improving treatment precision [200]. Second, off-target effects should be monitored using techniques such as single-cell metabolic flow tests, resulting in perfect delivery. Similarly, for mdivi-1, future studies should employ more rigorous approaches to link our aforementioned observations to Drp-1 inhibition, including Drp1 gene knockout/knockdown, the use of newer, more selective Drp1 inhibitors, and in vivo pharmacokinetic studies. Finally, multi-cycle, multi-species chronic toxicity trials are used to evaluate safety and enable precision treatment based on TC subtype, offering new and effective treatment schemes for patients with ATC, MTC, locally advanced-stage PTC, and targeted drug-resistant PTC.

Most current research is still confined to preclinical models, lacking systematic clinical studies to validate the clinical relevance of these targets. To compensate for existing deficiencies, organoids that cover multiple TC subtypes and humanized animal models should be developed. Furthermore, the mechanisms underlying the interaction between mitochondrial dysfunction and the tumor immune microenvironment are not yet fully elucidated, and how mitochondrial targeting influences anti-tumor immune responses requires further investigation. Future directions in TC therapy will focus on the precision regulation of mitochondrial function. To achieve this, multidimensional histological data, innovative bioengineering technologies, and individualized treatment strategies will be combined to create tailored treatment regimens for patient-specific mitochondrial abnormalities. With the continuous development of this field, a comprehensive intervention model that targets mitochondrial dysfunction—which includes metabolic modulation, mitochondrial transplantation strategies, and the synergistic application of targeted drugs—is anticipated to result in a groundbreaking advancement in TC therapy.

## Figures and Tables

**Figure 1 biomolecules-15-01292-f001:**
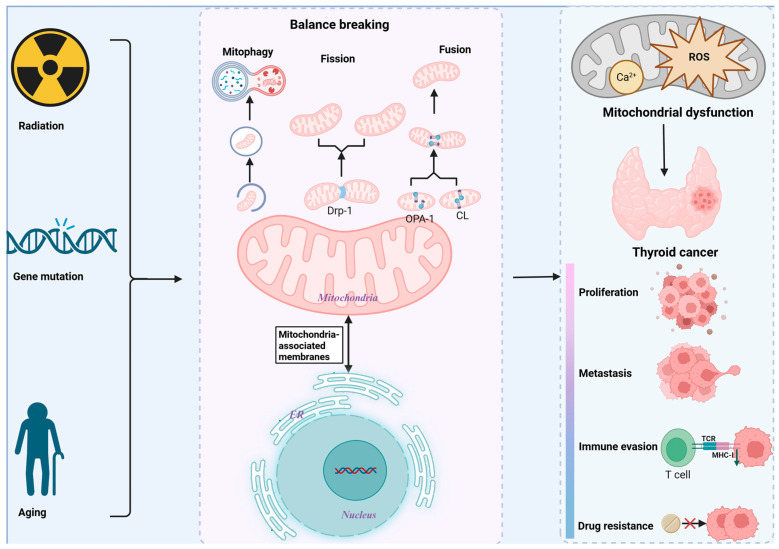
MQC imbalance and malignant progression of TC. Several factors influence and contribute to MQC imbalance, leading to mitochondrial dysfunction, TC proliferation, metastasis, and treatment resistance.

**Figure 2 biomolecules-15-01292-f002:**
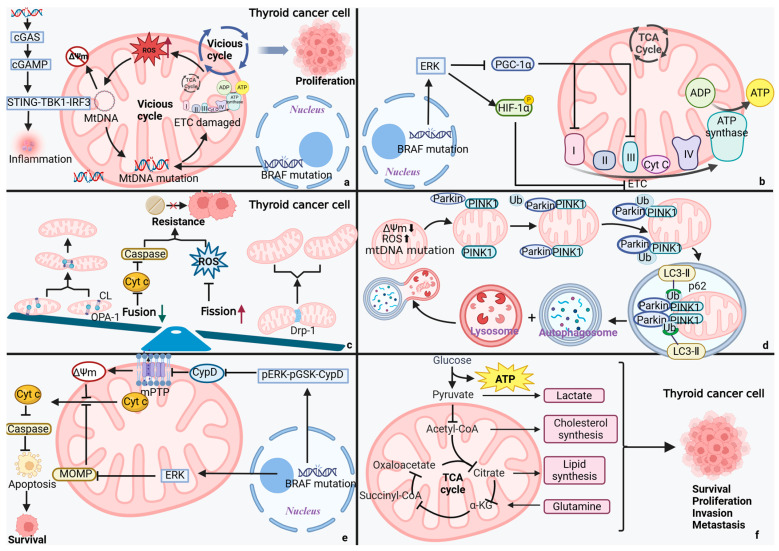
Mechanisms of mitochondrial dysfunction in the pathological process of TC. (**a**) MtDNA mutation that affects ETC causes ROS to be produced, which creates a vicious cycle that promotes TC proliferation. (b) ETC damage leads to impaired energy metabolism. (**c**) Disturbed mitochondrial dynamics (decreased mitochondrial fusion and increased mitochondrial fission) promotes TC resistance. (**d**) Abnormalities of mitochondrial autophagy. (**e**) Reduced mitochondrial permeability prevents TC cells from dying. (**f**) Metabolic reprogramming, which fosters TC invasion, metastasis, and proliferation. In the figure, red arrows indicate up-regulated components, functions, or metabolic patterns, and green arrows indicate down-regulated components, functions, or metabolic patterns.

**Figure 3 biomolecules-15-01292-f003:**
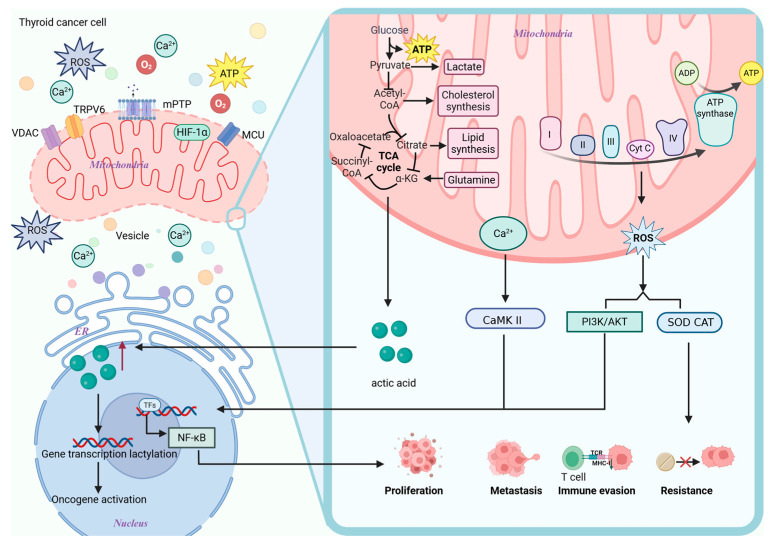
MIPS in TC. In TC, mitochondria serve as the cell’s multipurpose information processors, taking in, combining, and producing signals that support organismal and cellular adaptability. In the figure, red arrows indicate up-regulated components, functions, or metabolic patterns, and green arrows indicate down-regulated components, functions, or metabolic patterns.

**Figure 4 biomolecules-15-01292-f004:**
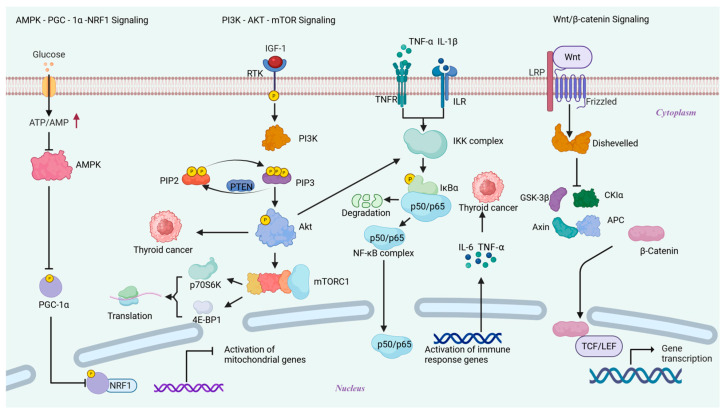
Cellular signaling pathways converge toward mitochondria in TC. Cellular homeostasis and the stress response are primarily influenced by various cellular signals gathered on mitochondria in TC, which aids in the development of TC. In the figure, red arrows indicate up-regulated components, functions, or metabolic patterns, and green arrows indicate down-regulated components, functions, or metabolic patterns.

**Figure 5 biomolecules-15-01292-f005:**
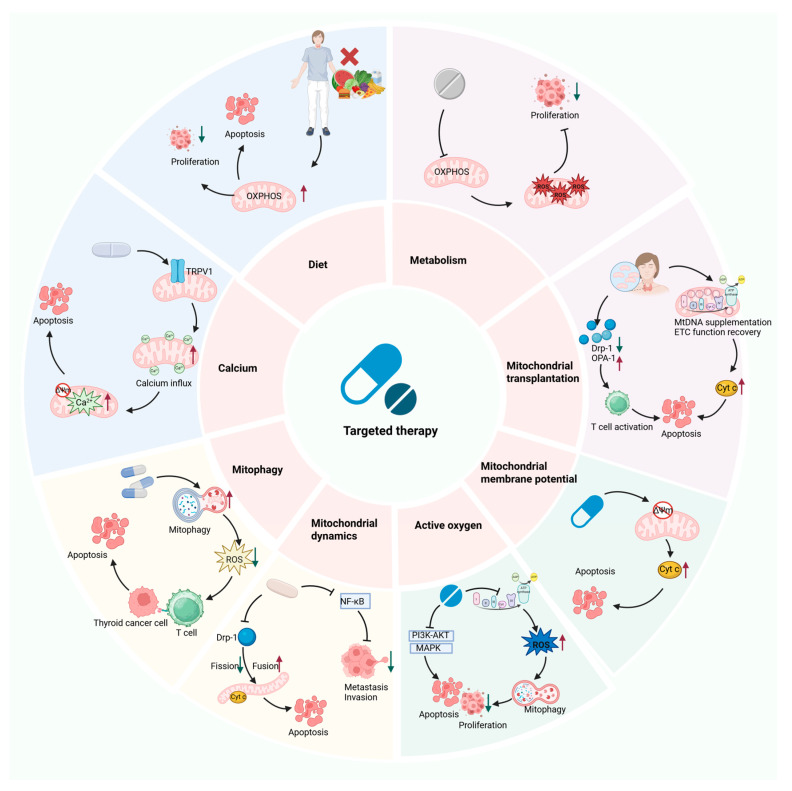
Therapeutic applications of mitochondrial transplantation and drugs targeting mitochondrial dysfunction. In the figure, red arrows indicate up-regulated components, functions, or metabolic patterns, and green arrows indicate down-regulated components, functions, or metabolic patterns.

**Table 1 biomolecules-15-01292-t001:** Mitochondrial dysfunction in TC and its specific impact on TC.

Type of Dysfunction	Specificities	Impact on TC	References
MtDNA mutation	Increased mtDNA mutations and copy number	1. Inhibits OXPHOS; increases ROS; induces metabolic reprogramming. 2. Promotes TC occurrence, proliferation, invasion, and metastasis.	[43,44,45]
Mitochondrial dynamics	Increased mitochondrial fission; Decreased mitochondrial fusion	1. Inhibits OXPHOS; promotes EMT. 2. Promotes TC proliferation, invasion, and metastasis; poor prognosis.	[46,47,48]
Mitochondrial autophagy	Downregulation of Parkin	1. Increases ROS; activates NF-κB. 2. Promotes metastasis, invasion, and drug resistance in thyroid Hürthle cell tumors.	[49]
Mitochondrial permeability	Decreased mitochondrial permeability	Inhibits apoptosis and pyroptosis in TC.	[50]
Disorders of mitochondrial metabolism	Promotes aerobic glycolysis and inhibits OXPHOS; Enhances glutamate, serine, and glycine metabolism; Increases fatty acid and cholesterol synthesis and uptake	Promotes TC survival, proliferation, invasion, and metastasis; poor prognosis.	[19,23,51,52,53,54]

**Table 2 biomolecules-15-01292-t002:** Preclinical trial drugs targeting mitochondrial dysfunction in TC.

Mitochondrial Dysfunction	Medicine	TC Subtype Tested	Studied Model	Primary Outcomes	Secondary Outcomes	Evidence Level	References
Mitochondrial autophagy	Curcumin	PTC, FTC and ATC	In vitro cell model (Nthy-ori-3.1, BCPAP, FTC133, and 8505C cells)	Decreased proliferation, and induced apoptosis in PTC, FTC, and ATC cells	Synergizes with RAI to kill TC cells	V	[125]
Mitochondrial fission	Mdivi-1	PTC and ATC	In vitro cell model (K1 cells and 8505C cells)	Decreased proliferation, invasion, and induced apoptosis in PTC and ATC cells	Reversed EMT by inhibiting the NF-κB pathway	V	[146]
Mitochondrial membrane potential	Myricetin	PTC	In vitro cell model (SNU-790 cells)	Decreased proliferation and induced apoptosis in PTC cells	Induces a decrease in mitochondrial membrane potential	V	[147]
Oxidative stress	Alantolactone	ATC	In vitro cell model (KHM-5M, KMH-2, C643CAL-62, and 8505C cells) and xenograft mouse model	Decreased proliferation and induced apoptosis in ATC cells	-	V	[148]
	Shikonin	PTC, FTC, and ATC	In vitro cell model (C643, 8305c, K1, BCPAP, TPC-1, IHH4, FTC133, HTori-3, and primaryTC cells) and xenograft mouse model	Decreased growth, migration, invasion, and induced apoptosis in TC cells	Reversed EMT by inhibiting multiple targets (e.g., Mdm2, Slug, MMPs)	V	[149]

**Table 3 biomolecules-15-01292-t003:** Clinical trial drugs targeting mitochondrial dysfunction in TC.

Mitochondrial Dysfunction	Medicine	Test Stage	Studied Model/Population	Experimental Group Sample Size	Control Group Sample Size	Outcomes	Adverse Reaction	NCT Identifier	Evidence Level	References
Mitochondrial metabolism	Tigecycline	Phase I trial (Single-Arm Trial)	Relapsed and refractory acute myeloid leukemia patients	27	-	Primary outcomes: maximal tolerated dose was 300 mg/day; Secondary outcomes: the t1/2 becomes shorter	The 300 mg/day dose was well-tolerated	NCT 01332786 *	IV	[150]
Mitochondrial fission	Ruxolitinib	Phase II trial (Single-Arm Trial)	Chronic neutrophilic leukemia and atypical chronic myeloid leukemia patients	44	-	Primary outcomes: ORR 32%; Secondary outcomes: CNL ORR 58%, aCML ORR 8%	Anemia (34%),Thrombocytopenia (14%)	NCT 02092324 *	II	[151]
Calcium regulation	Capsaicin	Phase III trial (Randomized Controlled Trial)	Patients with painful diabetic peripheral neuropathy	186	183	Primary outcomes: significant reduction in average daily pain; Secondary outcomes: proportion of responders ≥30% and ≥50% pain reduction	Application site pain (33.9%)	NCT 01533428 *	I	[152]
Oxidative stress	Vitamin C	Phase III trial (Randomized Controlled Trial)	Metastatic colorectal cancer patients	221	221	Primary outcomes: experimental group PFS 8.6 months, control group 8.3 months, HR 0.86; Secondary outcomes: experimental group and control group (ORR, 44.3% vs. 42.1%; OS, 20.7 vs. 19.7 months)	Neutropenia (14.9% vs. 15.4%), anemia (5.0% vs. 2.3%)	NCT 03146962 *	I	[153]
Mitochondrial membrane potential	Niclosamide	Phase Ib trial (Single-Arm Trial)	Castration-resistant prostate cancer patients	9	-	Primary outcomes: the maximal tolerated dose was 1200 mg, given three times a day; Secondary outcomes: 5 patients had ≥50% PSA response to treatment, and 2 of these patients had complete PSA response	The 1200 mg/day dose was well-tolerated	NCT 02807805 *	IV	[154]

Table note: The asterisk (*) indicates that the date of access for all URLs was 25 August 2025.

## Data Availability

No new data were created or analyzed in this study.

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
