# Peer review of "The Emerging Role of Mitochondrial Dysfunction in Thyroid Cancer: Mediating Tumor Progression, Drug Resistance, and Reshaping of the Immune Microenvironment"

_biomolecules, 2025, doi:10.3390/biom15091292_

Round 1
Reviewer 1 Report
Comments and Suggestions for Authors
Zhang et al. provides an insightful review that emphasizes the emerging role of mitochondrial dysfunction in thyroid cancer.
The manuscript is nicely written. I have a few comments for polishing the manuscript.
- The authors should address the crosstalk between thyroid cancer stem cells and alterations in mitochondrial functions.
- The authors should address the gaps that still need to be done in the area of research.
- The authors should highlight the importance of mitochondrial function in a broader manner in the Introduction section and how it impacts several other cancers. Like one recent evidence highlights its role in SCLC, https://doi.org/10.1158/1055-9965.EPI-19-0221 and https://doi.org/10.1016/j.isci.2025.112219 . The authors should consider the impact of mitochondrial dysfunction in relation to these and various other cancers.
Reviewer 2 Report
Comments and Suggestions for Authors
This manuscript provides a broad and up-to-date overview of the mechanisms of mitochondrial dysfunction in thyroid cancer, integrating aspects related to mtDNA, mitochondrial dynamics, autophagy, metabolism, and the tumor microenvironment. One of its main strengths is the solid bibliographic support, grounded in recent and relevant references. However, much of the text reads as an extensive literature compilation without offering an original critical analysis or formulating novel hypotheses. As a review, it should explicitly identify the principal knowledge gaps and open research questions in each subsection. Consequently, the following issues should be addressed and corrected before the manuscript can be considered for acceptance:
- The Introduction presents global statistics, but the authors should include authoritative sources (e.g., GLOBOCAN 2022 with DOI) and clearly distinguish incidence from mortality. In addition, the Introduction is overly narrative rather than quantitative; it should preferentially include concrete figures.
- In the paragraph beginning “Recent research has made it clear that mtDNA mutations…,” the authors state that mtDNA mutations are more often “passengers” than “drivers,” yet elsewhere in the same section they assert that mtDNA mutations are “significant drivers” in PTC. This contradiction must be reconciled and clarified.
- The example using 2-DG is useful, but the manuscript does not discuss toxicity or clinical feasibility; these issues should be addressed.
- Table 2 should include columns for “EVIDENCE LEVEL” and “TC SUBTYPE TESTED.” At present the table mixes preclinical and clinical data without distinction; it must indicate the trial stage (preclinical / in vitro / in vivo / Phase I / II / III), the studied population, N and primary/secondary outcomes. For example: Tigecycline is listed as “Phase I”, add the study N and endpoint; Mdivi-1 is marked preclinical, indicate the specific model.
- In the discussion of ruxolitinib, it is not clear whether the cited Phase II trial was performed in thyroid cancer or is an extrapolation from another indication. If it is extrapolated, state this explicitly.
- The Au-Ag@PDA nanoparticles example is mentioned without discussing limitations such as toxicity or biodistribution. Add a caveat on potential adverse effects and regulatory challenges.
- Standardize the use of the term “thyroid cancer” and the abbreviation “TC” throughout the manuscript.
- The Methods used to compile the review should be described: list consulted databases (PubMed/Embase/Scopus), search terms, dates, inclusion/exclusion criteria, and the number of articles selected.
- The text conflates effects of macroautophagy inhibitors (chloroquine, Lys05) with effects on mitophagy without clarifying specificity. Request that the authors: define the terms autophagy vs. mitophagy, review the cited experiments to confirm whether they specifically measure mitophagy (Parkin/PINK1 readouts, mito-Keima, etc.), and, if no TC-specific data exist, explicitly state that and avoid generalization.
- Wherever claims such as “increase,” “association,” or “poor prognosis” are made, quantitative values must be provided (e.g., % of samples with the mutation, hazard ratios, fold-change in Drp1/Mfn2 expression, ranges of mtDNA-CN) together with exact citations. Currently such assertions are mainly qualitative.
- Revise Table 1 (p.6) for format and typographical errors. For example, “PXPHOS” should be OXPHOS and “Decreasion” is a typo. Proofread the entire table for spelling and clarity.
- Maintain a single format for the BRAF mutation throughout the manuscript: either BRAF V600E or BRAFV600E (follow HGVS guidelines; BRAF V600E is preferred in running text). Homogenize usage everywhere.
- Figure 2 contains duplicated text in the legend. Remove repetition (“In the figure, red arrows indicate...”) and make the legend concise.
- The authors present Mdivi-1 as a “Drp-1 specific inhibitor” without discussing that its specificity and pharmacokinetics are debated in the literature (off-target effects and context-dependent activity). Add a critical discussion and propose more rigorous approaches (e.g., Drp1 genetic KD/KO, use of more selective inhibitors, or in vivo pharmacokinetic studies).
- Avoid strong claims unsupported by direct evidence in thyroid cancer. Sections (pp.18–20) extrapolate data from breast/prostate/ovarian cancers to TC. Add a critical subsection detailing technical feasibility, immunogenicity, delivery, storage, and risks of the proposed approaches, and explicitly state that there are no clinical trials in TC if that is the case.
- For each drug cited as having a clinical trial, provide the NCT identifier, trial phase, number of patients, and endpoints. Example: Niclosamide is cited with NCT02519582 for colorectal cancer, explicitly state that the cited NCT is for CRC and not for TC.
- The manuscript repeats core concepts (ROS, OXPHOS, EMT) in multiple sections. Consolidate basic explanations into a single “Key Concepts” section and refer back to it to improve concision.
Reviewer 3 Report
Comments and Suggestions for Authors
The review by Zhang et al aims to elucidate the molecular mechanisms by which mitochondrial dysfunction contributes to the development of thyroid cancer (TC) and to provide innovative intervention strategies for clinical treatment.
Overall, this review is very informative but does not always focus on mitochondria or mitochondrial dysfunction in thyroid cancer.
What I am missing is an introduction of the types of cells found in thyroid glands, the thyroid-hormone producing follicular cells and the parafollicular cells (C cells), which regulate calcium levels in the blood and help to lower calcium when it is too high. Missing is also what is special about mitochondria in these cell types and how the mitochondrial function differs in these cell types. Please add.
Abbreviation: Not all abbreviations are defined at their first mentioning; please change. I like the table of abbreviations but wish they would be alphabetically organized. In addition, there are unnecessary abbreviations like PPP that are not used after their initial definition.
Page 9, chapter 3.5: poorly differentiated TC and ATC: the change from a “normal” cell to a cancer cell often includes a de-differentiation.
Page 13, chapter 4.2: Beside cytosolic sources of ROS, the mitochondrial electron transport chain complexes I and III are known producers of ROS and should be discussed in this review.
Page 17: Mdivi-1 is not only an inhibitor of the mitochondrial fission protein Drp-1, but also an inhibitor of complex 1 of the electron transport chain (see for example Bordt et al in Developmental Cell, Volume 40, Issue 6, 27 March 2017, Pages 583-594.e6). Inhibition of electron transport chain activity would favor glycolytic energy generation and with that the growth of cancer cells Please revise this paragraph.
Paragraphs 6.3-6.7: Please revise these paragraphs by adding how the discussed treatment options could affect thyroid cancer.
Typos:
Page 4, chapter 3: “Novel opinions”; maybe options or opportunities were meant?
Page 5, chapter 3.1: “in the high cellular subtype of PTC”; something went wrong in this sentence.
Page 17: “cance” instead of cancer; directly before the reference 134. Same paragraph: “futur” instead of future directly before the reference 135.
Round 2
Reviewer 1 Report
Comments and Suggestions for Authors
Zhang et al. provides an insightful review that emphasizes the emerging role of mitochondrial dysfunction in thyroid cancer.
The authors have addressed all the previous comments. Thus, the manuscript can be accepted in its present form.
Reviewer 2 Report
Comments and Suggestions for Authors
The manuscript was modified ascordingly and was clearly improved
Reviewer 3 Report
Comments and Suggestions for Authors
No further comments.